# Real-Time Execution of Action Chunking Flow Policies

**Kevin Black**[1,2,]    **Manuel Y. Galliker**[1]    **Sergey Levine**[1,2]

[1]Physical Intelligence    [2]UC Berkeley

{kevin,manuel,sergey}@physicalintelligence.company

## Abstract

Modern AI systems, especially those interacting with the physical world, increasingly require real-time performance. However, the high latency of state-of-the-art generalist models, including recent vision-language-action models (VLAs), poses a significant challenge. While action chunking has enabled temporal consistency in high-frequency control tasks, it does not fully address the latency problem, leading to pauses or out-of-distribution jerky movements at chunk boundaries. This paper presents a novel inference-time algorithm that enables smooth asynchronous execution of action chunking policies. Our method, real-time chunking (RTC), is applicable to any diffusion- or flow-based VLA out of the box with no re-training. It generates the next action chunk while executing the current one, "freezing" actions guaranteed to execute and "inpainting" the rest. To test RTC, we introduce a new benchmark of 12 highly dynamic tasks in the Kinetix simulator, as well as evaluate 6 challenging real-world bimanual manipulation tasks. Results demonstrate that RTC is fast, performant, and uniquely robust to inference delay, significantly improving task throughput and enabling high success rates in precise tasks—such as lighting a match—even in the presence of significant latency. See `https://pi.website/research/real_time_chunking` for videos.

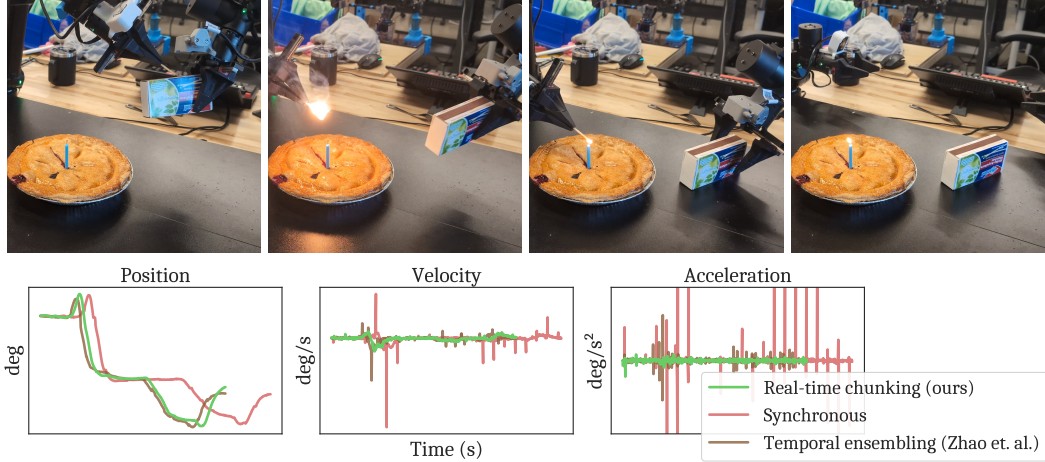

Figure 1: **Top:** Real-time chunking (RTC) enables the robot to perform highly dexterous and dynamic tasks, such as lighting a match—even in the presence of inference delays in excess of 300 milliseconds, corresponding to more than 30% of the model's prediction horizon. **Bottom:** RTC performs the same robot motion 20% faster than synchronous inference [5, 30, 8, 24, 31, 59], and smoother than all competing methods, including temporal ensembling [68]. The shown positions, velocities, and accelerations correspond to the shoulder joint of one arm, and are taken from the first 10 seconds of a real autonomous match-lighting rollout.

39th Conference on Neural Information Processing Systems (NeurIPS 2025).

# 1   Introduction

As AI systems have become more capable, they have also interacted more and more directly with their environment. Whether they're executing terminal commands [45], playing Pokémon on livestream [20], or browsing the web on your behalf [65], recent advances—driven primarily by large-scale deep learning—have enabled these systems to increasingly *control*, rather than merely *process*, the vast heterogeneity of the outside world. Embodied agents, where machine learning models directly control real, physical constructs, are perhaps the quintessential example. The same advances fueling agentic language and vision models are also making great strides in physical intelligence on platforms ranging from humanoid robots [4] to autonomous cars [60].

Cyber-physical systems, unlike chatbots and image generators, always operate in *real time*. While a robot is "thinking", the world around it evolves according to physical laws. Thus, delays between inputs and outputs have a tangible impact on performance. For a language model, the difference between fast and slow generation is a satisfied or annoyed user; for a robot action model, on the other hand, it could be the difference between a robot handing you a hot coffee or spilling it in your lap.

Unfortunately, the effectiveness of modern large-scale machine learning comes with high latency as an unavoidable side effect. Large language models (LLMs), vision-language models (VLMs), and vision-language-action models (VLAs)—the last referring to a class of models designed for visuomotor control—have billions of parameters [8, 30, 5, 4, 58]. These models are not only slow to run, but also require heavy-duty hardware that is difficult to attach to edge devices such as mobile robots, adding even more overhead for remote inference. Edge hardware will improve over time, but as robot datasets grow in size, so will the best VLAs [28].

Thus, applying large models to real-time control problems effectively will require some form of asynchronicity: that is, a model must think about its future actions while executing a previous one. Action chunking [68, 33, 11], where a model outputs and executes a sequence of multiple actions for each inference call, presents a partial solution. Although action chunking has already achieved many state-of-the-art results in dexterous manipulation [5, 4, 58], it still suffers from the latency problem. Chunking sacrifices the reactivity of a system to external stimuli and also introduces discontinuities in the transition points between chunks, as adjacent chunks may jump between different modes (or "strategies") from the learned action distribution. Such anomalies are especially harmful to learning-based systems, as they produce a distribution shift in dynamics that the model is likely not equipped to handle. Naive smoothing strategies, such as averaging multiple predictions together [68], are not guaranteed to produce valid actions and may only make matters worse (e.g., see Figure 2).

A good real-time system must produce a consistent and continuous control signal, incorporating the latest observations without perturbing the environment's natural dynamics or the model's ability to produce correct actions. In this work, we present **real-time chunking (RTC)**, which poses asynchronous action chunking as an inpainting problem. Our algorithm generates the next action chunk while executing the previous one, freezing the actions that are guaranteed to be executed (due to inference delay) and "inpainting" the rest. It is applicable to any diffusion- [22] or flow-based [36] VLA, and operates purely at inference time, requiring no changes to existing training recipes.

Our contributions are as follows. First, we present a novel system for asynchronous, real-time inference of action chunking diffusion- or flow-based policies for continuous control. Since standard simulation benchmarks are quasi-static—and have mostly been saturated with pseudo open-loop inference strategies [11]—we devise a new benchmark based on the Kinetix simulator [43] consisting of 12 highly dynamic manipulation and locomotion tasks. In the real world, we evaluate RTC on 6 challenging bimanual manipulation tasks using the $\pi_{0.5}$ VLA [24] as the base policy. Across both simulation and the real world, we demonstrate that RTC is fast and performant; it is uniquely robust to inference latency, even in highly precise tasks such as lighting a match (Figure 1), and it achieves greatly improved task throughput on all real tasks.

# 2   Preliminaries and Motivation

We begin with an action chunking policy denoted by $\pi(\mathbf{A}_t|\mathbf{o}_t)$, where $\mathbf{A}_t = [\mathbf{a}_t, \mathbf{a}_{t+1}, ..., \mathbf{a}_{t+H-1}]$ is a chunk of future actions, $\mathbf{o}_t$ is an observation, and $t$ indicates a controller timestep. We call $H$ the *prediction horizon*. When action chunking policies are rolled out, only the first $s \leq H$ actions from each chunk are executed. We call $s$ the *execution horizon*; often it is shorter than the

prediction horizon, but still much greater than 1 (e.g., $s \approx H/2$ [11, 5, 24]). Chunked execution ensures temporal consistency at the expense of reactivity. A long execution horizon reduces a policy's responsiveness to new information, while a short one increases the likelihood of mode-jumping, jerky behavior resulting from discontinuities between chunks.

In this paper, we consider policies trained with conditional flow matching [36], though our method can also be used with diffusion policies by converting them to flow policies at inference time [48, 18]. To generate an action chunk from a flow policy, random noise $\mathbf{A}_t^0$ is first sampled from a standard Gaussian, and then the flow's velocity field, $\mathbf{v}_\pi$ (a learned neural network) is integrated from $\tau = 0$ to 1 using the update rule

$$\mathbf{A}_t^{\tau + \frac{1}{n}} = \mathbf{A}_t^\tau + \frac{1}{n}\mathbf{v}_\pi(\mathbf{A}_t^\tau, \mathbf{o}_t, \tau), \tag{1}$$

where $\tau \in [0, 1)$ denotes a flow matching timestep, and $n$ determines the number of denoising steps.

Now, let $\Delta t$ be sampling period of the controller, i.e., the duration of a controller timestep, and let $\delta$ be the time it takes for the policy to generate an action chunk. We say that a system is *real-time* if it is guaranteed to produce a response (in our case: $\mathbf{a}_t$) to an event (receiving $\mathbf{o}_t$) within a fixed time constraint ($\Delta t$). If $\delta \leq \Delta t$, then meeting the real-time constraint is trivial, since an entire chunk can be generated between two controller timesteps. However, this is near impossible to achieve with modern VLAs. For example, with an RTX 4090 GPU, the 3 billion parameter $\pi_0$ VLA spends 46ms on the KV cache prefill alone, before any denoising steps [5], and targets a 50Hz control frequency ($\Delta t = 20$ms). Run in remote inference for mobile manipulation, $\pi_0$ lists 13ms of network latency, in perfect conditions with a wired connection. In a more realistic setting, the network overhead alone could easily exceed 20ms. Kim et al. [31], who optimize the 7B OpenVLA model [30] specifically for inference speed, achieve no better than 321ms of latency on a server-grade A100 GPU.

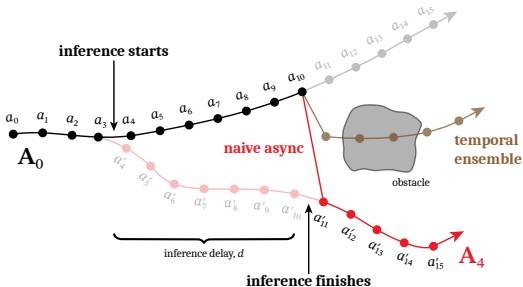

Figure 2: An illustration of a typical bifurcation between consecutive chunks. Inference is started between timesteps 3 and 4. The original chunk that was executing, $\{a_t\}$ (black), had planned to go above the obstacle while the newly generated chunk $\{a_t'\}$ (red) goes below the obstacle. However, $\{a_t'\}$ is not available until $d = 7$ steps later. A naive asynchronous algorithm might jump from $a_{10}$ to $a_{11}'$, inducing a very high, out-of-distribution acceleration. Temporal ensembling [68], i.e., interpolating between chunks, reduces the acceleration but produces poor actions.

Naive synchronous inference, the default in many prior works [5, 30, 8, 24, 31, 59], simply starts inference at the end of the execution horizon and waits while the policy generates the next chunk. When $\delta > \Delta t$, this introduces visible pauses between chunks that not only slow down execution but also change the dynamics of the robot, introducing distribution shift between training and evaluation. To develop a real-time strategy, we must first introduce *asynchronous* inference, where inference is started early and happens concurrently with execution.

We define $d := \lfloor \delta / \Delta t \rfloor$ and call this quantity the *inference delay*, corresponding to number of controller timesteps between when $\mathbf{o}_t$ is received and when $\mathbf{A}_t$ is available.[1] Let $\mathbf{a}_{t'|t}$ denote the $(t' - t)$-th action of chunk $\mathbf{A}_t$, generated from observing $\mathbf{o}_t$. If $\mathbf{A}_0$ is currently executing, and we desire an execution horizon of $s$, then an asynchronous algorithm must start inference at $s - d$. So long as $d \leq H - s$, then this strategy will satisfy the real-time constraint and guarantee that an action is always available when it is needed. However, since the policy cannot know what will happen between steps $s - d$ and $s$ while generating $\mathbf{A}_{s-d}$, the transition point between $\mathbf{a}_{s-1|0}$ and $\mathbf{a}_{s|s-d}$ may be arbitrarily discontinuous and out-of-distribution. Similar to a too-short execution horizon, this strategy leads to jerky behavior that is worsened dramatically with higher delays; see Figure 2.

---

[1]For simplicity, we do not consider delays or synchronization issues at the sub-timestep level; we assume that the environment or lower-level controller provides $\mathbf{o}_t$ at the same instant that $\mathbf{a}_{t-1}$ is consumed.

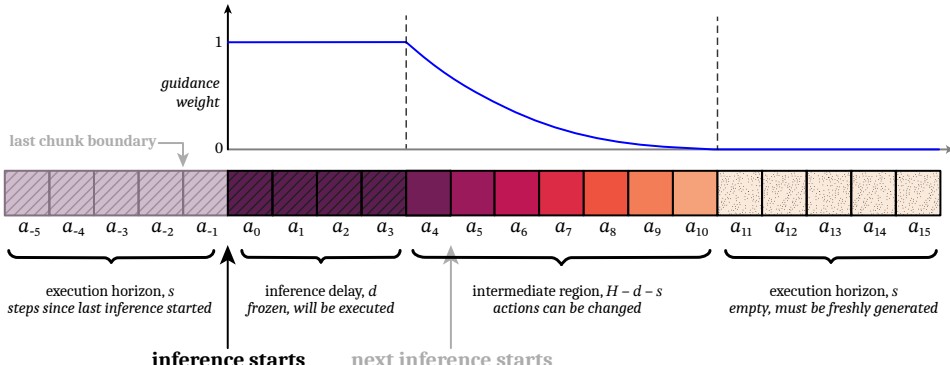

Figure 3: A diagram illustrating how action generation attends to the previous action chunk in real-time chunking. If inference starts after the execution of $a_{-1}$ and the inference delay is $d = 4$, then the newly generated chunk will not be available until after $a_3$ is consumed. Therefore, $a_{0:3}$ are "frozen" and are attended to with a full guidance weight of 1. In the intermediate region, $a_{4:10}$, actions from the previous chunk are available but may be updated, since inference will have finished before $a_4$ is needed. This region is attended to with an exponentially decreasing guidance weight. Finally, the last $s = 5$ actions are beyond the end of the previous chunk, and need to be freshly generated. The execution horizon, $s$, is a hyperparameter constrained by $d \leq s \leq H - d$.

## 3 Real-Time Chunking via Inpainting

The key challenge in real-time execution is to maintain continuity between chunks. By the time a new chunk is available, the previous one has already been executed partway, and therefore the new chunk must be "compatible" with the previous one. At the same time, the new chunk should still incorporate new observations, so that the policy does not lose the ability to react and make corrections.

Our key insight is to pose real-time chunking as an inpainting problem. To make the new chunk "compatible", we must use the overlapping timesteps where we have access to the remaining actions of the previous chunk. The first $d$ actions from the new chunk cannot be used, since those timesteps will have already passed by the time the new chunk becomes available. Thus, it makes sense to "freeze" those actions to the values that we know *will* be executed; our goal is then to fill in the remainder of the new chunk in a way that is consistent with this frozen prefix (see Figure 3), much like inpainting a section of an image that has been removed. We describe this basic inpainting principle in Sec. 3.1. In Sec. 3.2, we introduce a *soft masking* extension that is critical for full cross-chunk continuity; finally, we describe our full real-time chunking system in Sec. 3.3.

### 3.1 Inference-Time Inpainting with Flow Matching

Inpainting is a known strength of iterative denoising frameworks such as diffusion and flow matching. We build on the training-free image inpainting algorithm from Pokle et al. [48], which is itself based on pseudoinverse guidance (ΠGDM; [55]). The algorithm operates by adding a gradient-based guidance term to the learned velocity field $\mathbf{v}$ at each denoising step (Equation 1) that encourages the final generation to match some target value, $\mathbf{Y}$, which is a corrupted version of the desired result. In the case of image inpainting, the corruption operator is masking, $\mathbf{Y}$ is the masked image, and the desired result is a full image consistent with $\mathbf{Y}$ in the non-masked areas. The ΠGDM gradient correction, specialized to our setting, is given by

$$\mathbf{v}_{\Pi\text{GDM}}(\mathbf{A}_t^\tau, \mathbf{o}_t, \tau) = \mathbf{v}(\mathbf{A}_t^\tau, \mathbf{o}_t, \tau) + \min\left(\beta, \frac{1-\tau}{\tau \cdot r_\tau^2}\right)\left(\mathbf{Y} - \widehat{\mathbf{A}_t^1}\right)^\top \text{diag}(\mathbf{W})\frac{\partial \widehat{\mathbf{A}_t^1}}{\partial \mathbf{A}_t^\tau} \quad (2)$$

$$\text{where } \widehat{\mathbf{A}_t^1} = \mathbf{A}_t^\tau + (1-\tau)\mathbf{v}(\mathbf{A}_t^\tau, \mathbf{o}_t, \tau), \quad (3)$$

$$r_\tau^2 = \frac{(1-\tau)^2}{\tau^2 + (1-\tau)^2}. \quad (4)$$

$\widehat{\mathbf{A}_t^1}$ is an estimate of the final, fully denoised action chunk and $\mathbf{W}$ is the mask. We are abusing notation by treating $\mathbf{Y}$, $\mathbf{A}_t$, and $\mathbf{W}$ as vectors of dimension $HM$ where $M$ is the dimension of each action. Thus, the guidance term is a vector-Jacobian product and can be computed using

backpropagation. The guidance weight clipping, $\beta$, is our addition; we found that without it, the algorithm became unstable with the small number of denoising steps commonly used in control problems (see A.2 for an ablation).

## 3.2 Soft Masking for Improved Cross-Chunk Continuity

In practice, naively inpainting using only the first $d$ timesteps of the previous action chunk is often insufficient to ensure that the new chunk takes a consistent strategy, particularly when $d$ is small (e.g., see Figure 4). The $\Pi$GDM correction is not perfect, and a small $d$ leads to a weak guidance signal, which can allow for the new chunk to still switch strategies and cause discontinuities. Our solution, illustrated in Figure 3, is to give our policy more cross-chunk continuity by considering not just the first $d$ overlapping actions, but all $H - s$ overlapping actions. We do this via *soft masking*, setting $\mathbf{W}$ to real-valued weights rather than 1s and 0s. The first $d$ actions get a weight of 1; the last $s$ actions of the new chunk do not overlap with the previous chunk, so they get a weight of 0; the actions in between get weights that exponentially decay from 1 to 0, accounting for the fact that actions further in the future should be treated with more uncertainty. The resulting expression for $\mathbf{W}$ is given by

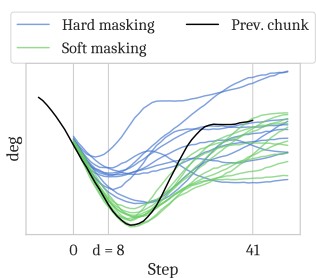

Figure 4: A comparison of naive inpainting (hard masking) and our proposed soft masking method: note that hard masking does not match the frozen region very well and produces faster changes in direction.

$$\mathbf{W}_i = \begin{cases} 1 & \text{if } i < d \\ c_i \frac{e^{c_i}-1}{e-1} & \text{if } d \le i < H - s \\ 0 & \text{if } i \ge H - s \end{cases} \quad \text{where } c_i = \frac{H - s - i}{H - s - d + 1}, \quad i \in \{0, \dots, H - 1\}. \quad (5)$$

Intuitively, $\mathbf{W}$ modulates the "attention" paid to each corresponding action from the previous chunk. See Appendix A.4 for a comparison between different decay schedules.

## 3.3 Real-Time Chunking

We present our full real-time chunking system in Algorithm 1 (complemented by Figure 3). The controller interfaces with our algorithm via GETACTION, which is called every $\Delta t$ to consume an action $\mathbf{a}_{t-1}$ and provide the next observation $\mathbf{o}_t$. The INFERENCELOOP runs in a background thread so that an action is always available. It forecasts the next delay, $d$, by keeping a buffer of past delays. The execution horizon, $s$, can change from chunk to chunk; the user provides a minimum desired horizon, $s_{\min}$, and the actual horizon for a given chunk is $\max(d, s_{\min})$ where $d$ is the delay encountered when computing the *next* chunk. Finally, the algorithm describes the inpainting with soft masking procedure in GUIDEDINFERENCE, which explicitly defines a denoising function (Eq. 3) and computes a vector-Jacobian product, which can be done with reverse-mode autodifferentiation [2].

---

**Algorithm 1** Real-Time Chunking

---

**Require:** flow policy $\pi$ with prediction horizon $H$, minimum execution horizon $s_{\min}$, mutex $\mathcal{M}$, condition variable $\mathcal{C}$ associated with $\mathcal{M}$, initial chunk $\mathbf{A}_{\text{init}}$, initial delay estimate $d_{\text{init}}$, delay buffer size $b$, number of denoising steps $n$, maximum guidance weight $\beta$

1: **procedure** INITIALIZESHAREDSTATE               ▷ Initialize mutex-protected shared variables
2:      $t = 0$; $\mathbf{A}_{\text{cur}} = \mathbf{A}_{\text{init}}$, $\mathbf{o}_{\text{cur}} = $ null

3: **function** GETACTION($\mathbf{o}_{\text{next}}$)                  ▷ Called at an interval of $\Delta t$ by controller
4:      **with** $\mathcal{M}$ acquired **do**
5:          $t = t + 1$
6:          $\mathbf{o}_{\text{cur}} = \mathbf{o}_{\text{next}}$
7:          notify $\mathcal{C}$
8:          **return** $\mathbf{A}_{\text{cur}}[t - 1]$

---

9: **procedure** INFERENCELOOP             ▷ Run inference in a looping background thread
10:      acquire $\mathcal{M}$
11:      $\mathcal{Q}$ = new Queue($[d_{\text{init}}]$, maxlen=$b$)      ▷ Holds a limited buffer of past inference delays
12:      **loop**
13:          wait on $\mathcal{C}$ until $t \geq s_{\min}$
14:          $s = t$      ▷ $s$ is the number of actions executed since last inference started
15:          $\mathbf{A}_{\text{prev}} = \mathbf{A}_{\text{cur}}[s, s+1, \ldots, H-1]$      ▷ Remove the $s$ actions that have already been executed
16:          $\mathbf{o} = \mathbf{o}_{\text{cur}}$
17:          $d = \max(\mathcal{Q})$      ▷ Estimate the next inference delay conservatively
18:          **with** $\mathcal{M}$ released **do**
19:              $\mathbf{A}_{\text{new}} = \text{GUIDEDINFERENCE}(\pi, \mathbf{o}, \mathbf{A}_{\text{prev}}, d, s)$
20:          $\mathbf{A}_{\text{cur}} = \mathbf{A}_{\text{new}}$      ▷ Swap to the new chunk as soon as it is available
21:          $t = t - s$      ▷ Reset $t$ so that it indexes into $\mathbf{A}_{\text{new}}$
22:          enqueue $t$ onto $\mathcal{Q}$      ▷ Record the observed delay

23: **function** GUIDEDINFERENCE$(\pi, \mathbf{o}, \mathbf{A}_{\text{prev}}, d, s)$
24:      compute $\mathbf{W}$ using Eq. 5; right-pad $\mathbf{A}_{\text{prev}}$ to length $H$; initialize $\mathbf{A}^0 \sim \mathcal{N}(\mathbf{0}, \mathbf{I})$
25:      **for** $\tau = 0$ to $1$ with step size $1/n$ **do**
26:          $f_{\widehat{\mathbf{A}^1}} = \mathbf{A}' \mapsto \mathbf{A}' + (1-\tau)\mathbf{v}_\pi(\mathbf{A}', \mathbf{o}, \tau)$      ▷ Define denoising function (Eq. 3)
27:          $\mathbf{e} = \left(\mathbf{A}_{\text{prev}} - f_{\widehat{\mathbf{A}^1}}(\mathbf{A}^\tau)\right)^\top \text{diag}(\mathbf{W})$      ▷ Weighted error term from Eq. 2
28:          $\mathbf{g} = \mathbf{e} \cdot \left.\frac{\partial f_{\widehat{\mathbf{A}^1}}}{\partial \mathbf{A}'}\right|_{\mathbf{A}'=\mathbf{A}^\tau}$      ▷ Compute vector-Jacobian product from Eq. 2 via autodiff
29:          $\mathbf{A}^{\tau+\frac{1}{n}} = \mathbf{A}^\tau + \frac{1}{n}\left(\mathbf{v}_\pi(\mathbf{A}^\tau, \mathbf{o}, \tau) + \min\left(\beta, \frac{1-\tau}{\tau \cdot r_\tau^2}\right)\mathbf{g}\right)$      ▷ Integration step (Eq. 1)
     **return** $\mathbf{A}^1$

## 4 Experiments

In our experiments, we aim to answer the following questions. First, how does RTC compare to existing methods in highly dynamic and stochastic environments, and under increasing inference delays? Second, how important is soft masking (Sec. 3.2) to RTC? Third, how does RTC affect the performance *and* speed of real-world dexterous robots?

We first evaluate RTC using a benchmark of 12 highly dynamic and stochastic environments in the Kinetix [43] simulator. We use this benchmark to compare the performance of RTC to other methods under simulated inference delays, as well as investigate the effect of soft masking. Then, using the $\pi_{0.5}$ VLA [24] as the base model, we evaluate the performance and speed of RTC on 6 challenging bimanual dexterous manipulation tasks, including 2 mobile manipulation tasks (see Figure **??**).

### 4.1 Simulated Benchmark

Most simulated imitation learning benchmarks are quasi-static, and standard chunked execution with a long enough execution horizon can achieve near-perfect success rates [11]. We instead create a benchmark of 12 dynamic tasks in Kinetix [43], which uses force-based control, so inference delay *necessitates* asynchronous execution (there is no concept of "holding position"). We select 10 existing environments and create 2 new ones such that all environments involve dynamic motions like throwing, catching, and balancing. To simulate imperfect actuation, we add Gaussian noise to the actions, making closed-loop corrections crucial for success.

**Setup.** To generate data for imitation learning, we first train expert policies using RPO [50] and a binary success reward. For each environment, we train 6 expert policies with different seeds and then generate a 1M transition dataset with a different policy selected each episode. We then train action chunking flow policies with a prediction horizon of $H = 8$ and a 4-layer MLP-Mixer [61] architecture for 32 epochs. We report binary success rates with 2048 rollouts per data point, and simulate delays between 0 (fully closed-loop) and 4 (the maximum supported when $H = 8$).

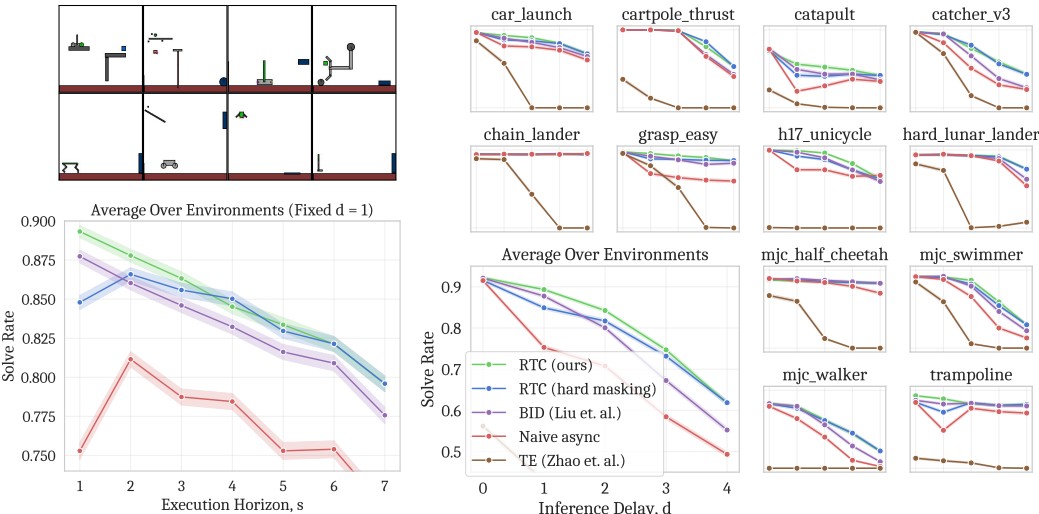

Figure 5: **Top left:** Kinetix environments; each involves getting a green object on the left to touch a blue one on the right. **Bottom left:** Execution horizon vs. solve rate with a fixed inference delay of 1. Only RTC and BID take full advantage of faster updates, showing strictly increasing performance with decreasing execution horizon. **Right:** Inference delay vs. solve rate with a fixed execution horizon of $s = \max(d, 1)$. RTC outperforms all baselines. Furthermore, soft masking (Sec. 3.2) improves performance at lower inference delays and execution horizons. Each data point represents 2048 trials, and 95% Wilson score intervals are shaded in.

**Baselines.** We compare against the following baselines:

- *Naive async.* This strategy does not pay attention to the previous action chunk at all when generating a new one, naively switching chunks as soon as the new one is ready.
- *Bidirectional decoding (BID; [39]).* This strategy uses rejection sampling to keep continuity across chunks. We use a batch size of $N = 32$, mode size of $K = 3$, and a checkpoint trained for 8 epochs as the weak policy.
- *Temporal ensembling (TE; [68]).* This strategy involves keeping a buffer of predicted action chunks and executing an average of all actions predicted for a particular timestep.

**Results.** Figure 5 shows the simulated results. In the delay plots (right): TE performs poorly across the board, even with an inference delay of $d = 0$, illustrating the multi-modality of our benchmark—averages of valid actions are not necessarily valid. RTC shows the most robustness to inference delays, outperforming BID, and the gap widens with increasing delay; note that BID uses significantly more compute than RTC by sampling batches of 64 action chunks, 32 from a strong model and 32 from a weak model. Additionally, we find that hard masking somewhat underperforms soft masking, particularly when $d$ is smaller, supporting our claims in Sec. 3.2. Finally, in the execution horizon plot (left), we find that thanks to its continuity across chunks, RTC is better able to take advantage of closed-loop corrections, always performing better with a decreasing execution horizon.

## 4.2 Real-World Results

Next, we deploy our full real-time chunking system to the real world. We use the $\pi_{0.5}$VLA [24] as our base policy, and evaluate RTC on a bimanual system with two 6-DoF arms and parallel jaw grippers. Unlike our simulated benchmark, the robots use position control, and so synchronous inference—stopping between chunks—is a reasonable default strategy, used in many prior works [5, 24, 31, 47]. Our goal is to improve upon synchronous inference in a combination of both performance *and* speed.

**Setup.** We use $\pi_{0.5}$ ($H = 50$, $\Delta t = 20$ms) with $n = 5$ denoising steps, giving a model latency of 76ms for the baselines and 97ms for RTC. We use remote inference over LAN, which adds 10-20ms of latency, giving a starting inference delay around $d \approx 6$ for RTC. However, we would like to understand how the system behaves with higher inference latencies, simulating, e.g., scaling up the model size or running inference on a distant cloud server. Thus, we also evaluate all methods with +100ms and +200ms of injected latency, corresponding to $d \approx 11$ and $d \approx 16$, respectively.

**Tasks and scoring.** Each episode gets an integer score corresponding to how many substeps of the task it completed successfully. We evaluate the following tasks:

- *Light candle (5 steps, 40s cutoff).* Pick up a match and matchbox, strike the match, use it to light a candle, and drop it in a bowl.
- *Plug ethernet (6 steps, 120s cutoff).* Pick up the end of an ethernet cable, reorient it, plug it into a server rack, and repeat the process for the other end.
- *Make bed, mobile (3 steps, 200s cutoff).* Move the corner of a blanket and 2 pillows from the foot to the head of a bed.
- *Shirt folding (1 step, 300s cutoff).* Fold a shirt from a flattened position.
- *Batch folding (4 steps, 300s cutoff).* Take a varied, crumpled clothing item out of a bin, flatten it, fold it, then place it neatly on a pile.
- *Dishes in sink, mobile (8 steps, 300s cutoff).* Move 4 varied items from a counter into a sink.

See the accompanying blog post for videos of each task. We evaluate each task and method for 10 trials for a total of 480 episodes, adding up to 28 hours of pure robot execution time. We also post-hoc annotate the score for each episode and the timestamp at which each step is achieved.

**Baselines.** We compare to the following baselines:
- *Synchronous.* This corresponds to the default inference strategy in prior work [5, 24, 31, 47], which executes $s = 25$ actions and then pauses while the new chunk is generated.
- *TE, sparse.* This is similar to *naive async* in our simulated results; it executes $s = 25$ actions at a time while computing the next chunk in parallel. We found it significantly reduced jerkiness to also apply TE, even though only the first $H - s - 2d$ executed steps of each chunk have overlapping actions to ensemble.
- *TE, dense.* This strategy is the closest to the original TE in Zhao et al. [68]. We run inference as often as possible, resulting in $s = d$ for every chunk. This results in there always being at least 2 overlapping action chunks to ensemble, and often more.

We do not compare to BID [39] in the real world, as we found in simulation that it underperforms RTC while using significantly more compute—when applied to $\pi_{0.5}$ with a batch size of 16, BID has 2.3 times the latency of our method (see A.3 for latency measurements).

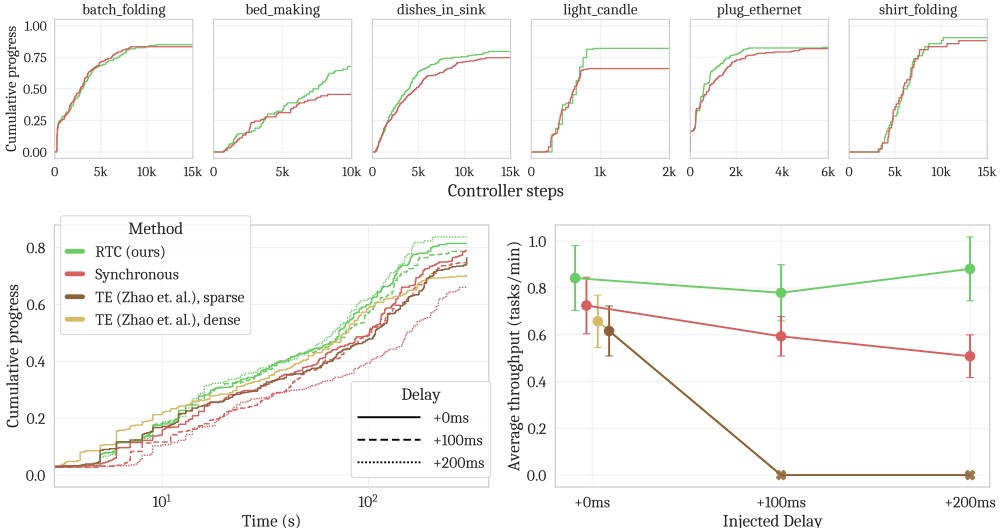

Figure 6: **Top:** Controller steps (equivalent to elapsed time with inference pauses removed multiplied by 50Hz) vs. cumulative progress for each task, aggregated across all delays. Progress is measured in discrete steps corresponding to the subsections of each task. **Left:** Time (including inference pauses) vs. cumulative progress aggregated across all tasks. The x-axis is log scale to better show progress during both short and long-horizon tasks. **Right:** Inference delay vs. average throughput, defined as the proportion of task completed divided by duration of episode averaged over episodes. Error bars are $\pm 1$ SEM. Average throughput gives a balanced view of both speed and performance for each method. Neither TE variant can run at +100 or +200ms of injected latency, causing such high oscillations that the robot's protective stop is triggered.

**Results.** We present the results in Figure 6. In average task throughput, a measurement of both speed and performance, RTC achieves the best score at all inference delays with a statistically significant result at +100 and +200ms. RTC is completely robust to injected delay, showing no degradation,

whereas synchronous degrades linearly and both TE variants do not run at all due to causing such high oscillations that the robot's protective stop is triggered (see videos). Inspecting the per-task results (Figure 5, top), we can conclude that RTC helps with more than just execution speed: it completes tasks faster than synchronous inference **even when inference pauses are removed**. All tasks, except for *light candle*, allow for retrying until the time limit (and $\pi_{0.5}$ does, in general, exhibit robust retrying behavior). Even though synchronous inference often reaches a similar final score, RTC often completes more of the task earlier in the episode, reflecting fewer mistakes and less retrying. In *light candle*, the most precision-sensitive task—and also the only one without retrying—RTC shows a large advantage in final score, reflecting a higher overall success rate. Interestingly, the same is true in *bed making*, even though that task does elicit retrying. The policy particularly struggles to manipulate the pillows, and *bed making* is the hardest task overall, which may be why RTC has a strong effect.

## 5   Related Work

**Action chunking, VLAs, and cascade control.** Inspired in part by human motor control [33], action chunking has recently emerged as the de facto standard in imitation learning for visuomotor control [68, 11]. Learning to generate action chunks from human data requires expressive statistical models, such as variational inference [68, 19], diffusion [11, 12, 69, 68, 46, 59], flow matching [5, 6], vector quantization [34, 3, 44], or byte-pair encoding [47]. Recently, some of these methods have been scaled to billions of parameters, giving rise to VLAs [7, 13, 30, 5, 71, 10, 9, 70, 24, 47, 37], a class of large models built on pre-trained vision-language model backbones. With the capacity to fit ever-growing robot datasets [13, 29, 62, 15, 41, 27], as well as Internet knowledge from vision-language pre-training, VLAs have achieved impressive results in generalizable robot manipulation. When applied to real-world robots, action chunking policies are often used in conjunction with a lower-level, higher-frequency control loop—such as a PID controller—which translates the outputs of the policy (e.g., joint positions) to hardware-specific control signals (e.g., joint torques). In these cases, action chunking policies can be viewed as a form of cascade control [14], with the learned policy acting the outermost control loop. However, this is not always the case: for example, our simulated experiments use learned policies that output torques and forces directly. As such, we defer any exploration of the intersection between cascade control theory and learned action chunking policies to future work.

**Reducing inference latency.** A natural approach to improve the real-time capabilities of a model is to simply speed it up. For instance, consistency policy [49] distills diffusion policies to elide expensive iterative denoising. Streaming diffusion policy [23] proposes an alternative training recipe that allows for very few denoising steps per controller timestep. Kim et al. [31] augment OpenVLA [30] with parallel decoding to elide expensive autoregressive decoding. More broadly, there is a rich literature on optimizing inference speed, both for diffusion models [52, 38, 56, 17] and large transformers in general [32, 25, 35]. Unfortunately, these directions cannot reduce inference cost below one forward pass. So long as this forward pass takes longer than the controller's sampling period, other methods will be needed for real-time execution.

**Inpainting and guidance.** There is a rich literature on image inpainting with pre-trained diffusion and flow models [48, 55, 40, 42]. In our work, we incorporate one such method [48] into our novel real-time execution framework with modifications (namely, soft masking and guidance weight clipping) that we find necessary for our setting. For sequential decision-making, Diffuser [26] pioneered diffusion-based inpainting for following state and action constraints in long-term planning, though their inpainting method is not guidance-based. (See Appendix A.4 for a comparison to the inpainting method from Diffuser applied to our setting.) Diffuser and other work [64, 1] have also guided diffusion models with value functions to solve reinforcement learning (RL) problems. Our work is distinct in that it is the first to apply either inpainting or guidance to real-time control.

**Real-time execution.** Real-time control has been studied long before the advent of VLAs. Similar to action chunking, model predictive control (MPC; [51]) generates plans over a receding time horizon; like our method, it parallelizes execution and computation, and uses the prior chunk to warm-start planning for the next. Though recent works combining learning methods with MPC have demonstrated real-time control capabilities in narrow domains [53, 21], they rely on explicit, hand-crafted dynamics models and cost functions. These methods are not applicable to our setting, which considers model-free imitation learning policies and tests them on unstructured, open-world manipulation tasks. Separately, in reinforcement learning, a variety of prior works have developed time-delayed decision-making methods [57, 16, 54, 63, 66, 67]. However, these approaches are not

always applicable to imitation learning, and none of them leverage action chunking. Most recently, hierarchical VLA designs [58, 4] have emerged where the model is split into a System 2 (high-level planning) and System 1 (low-level action generation) component. The System 2 component contains the bulk of the VLA's capacity and runs at a low frequency, while the System 1 component is lightweight and fast. This approach is orthogonal to ours, and comes with its own tradeoffs (e.g., limiting the size of the System 1 component and requiring its own training recipe).

**Bidirectional Decoding.** The most closely related prior work is Bidirectional Decoding (BID; [39]), which enables fully closed-loop control with pre-trained action chunking policies via rejection sampling. While Liu et al. [39] do not consider inference delay, the BID algorithm can be used to accomplish the same effect as our guidance-based inpainting. We compare to BID in our simulated benchmark, finding that it underperforms RTC while using significantly more compute.

# 6 Discussion and Future Work

Real-time chunking is an inference-time algorithm for asynchronous execution of action chunking policies that demonstrates speed and performance across simulation and real-world experiments, including under significant inference delays. However, this work is not without limitations: it adds significant computational overhead compared to methods that sample directly from the base policy, and it is applicable only to diffusion- and flow-based policies. Additionally, while our real-world experiments cover a variety of challenging manipulation tasks, there are more dynamic settings that could benefit even more from real-time execution. One example is legged locomotion, which is represented in our simulated benchmark but not our real-world results.

# 7 Acknowledgements

We thank Charles Xu and Kay Ke for designing the Ethernet plug-in task. We thank Brian Ichter for suggesting the cumulative progress plots and for later feedback on figures. We thank Dibya Ghosh for suggesting the throughput metric to measure a combination of speed and performance. We thank Ury Zhilinsky, Karan Dhabalia, Haohuan Wang, and Dibya Ghosh for help with training infrastructure; Noah Brown, Szymon Jakubczak, Adnan Esmail, Tim Jones, Mohith Mothukuri, James Darpinian, and James Tanner for robot infrastructure; Adrian Li-Bell for evaluation infrastructure; Anna Walling, Chelsea Finn, and Karol Hausman for robot, data and evaluation operations; and Michael Equi, Quan Vuong, and Jost Tobias Springenberg for training some of the $\pi_{0.5}$ policies used in the real-world experiments. We also thank Claudio Guglieri and Alex Krasikov for their help with visualizations for the blog post, and Jessica Dai for helpful copy editing of the paper manuscript. Finally, we are grateful to the whole team of robot operators at Physical Intelligence for their enormous contributions to running data collection and policy evaluations.

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

Figure 7: **Top left:** The graph of the value $\frac{1-\tau}{\tau \cdot r_\tau^2}$ from Eq. 2, which we clip at $\beta$. At $\tau = 0$, clipping is needed to make the value finite. With 5 denoising steps, if $\beta \geq 4.25$, the clipping only determines the guidance weight for the first step ($\tau = 0$). **Top right:** An ablation of $\beta$ in our simulated benchmark. Increasing $\beta$ provides no marginal benefit beyond $\beta = 5$. **Bottom left:** Example real robot action chunks generated from the same noise with 5 denoising steps ($n = 5$) and 100 denoising steps ($n = 100$), with lower opacities corresponding to higher guidance weight clipping ($\beta = \{5, 20, 50, 150\}$). With 5 denoising steps, the generated action chunks diverge when $\beta$ is too high. **Bottom right:** $\beta$ vs. maximum acceleration (second discrete difference) for a batch of 325 action chunks generated with $d = 15$ and $n = 5$. Higher $\beta$ leads to more jerkiness, a proxy for out-of-distribution actions.

# A    Appendices

## A.1    Broader Impacts

The goal of our work is to improve the speed and performance of learned policies for control tasks, and our experiments primarily deal with household robots. This technology has great potential to improve lives, e.g., by automating dangerous and difficult jobs, or assisting the disabled and elderly. Like any technology, it also has the potential for harm—e.g., in military applications, or by displacing physical labor.

## A.2    The Necessity of Guidance Weight Clipping ($\beta$)

In Section 3.1, we describe how we adapt the inpainting algorithm from Pokle et al. [48] and Song et al. [55] to our setting. One modification we make is to add a clipping value, $\beta$, which limits weight applied to the guidance term (Eq. 2), and is necessary to make the weight finite at $\tau = 0^2$. While image inpainting typically uses a high number of denoising steps (e.g., $n = 100$ in [48]), control problems often use very few steps (e.g., $n = 5$ in our experiments). In this case, we found that high guidance weights led to diverging action chunks, as shown in Figure 7, bottom left. Based on a simulated ablation (Figure 7, top right), we set $\beta$ to a conservative value of 5.

---

[2]An alternative approach to avoid the infinite weight at $\tau = 0$ is to start denoising from $\tau > 0$, used in [48], which we did not try.

## A.3 Latency Measurements

| Method | Latency |
|---|---|
| RTC (ours) | 97ms |
| BID with $N = 16$ (no forward model) | 115ms |
| BID with $N = 16$ (shared backbone) | 169ms |
| BID with $N = 16$ (full) | 223ms |
| Vanilla $\pi_{0.5}$ | 76ms |

Table 1: Latency measurements for various inference-time methods applied to $\pi_{0.5}$ [24]. Numbers include on-GPU neural network inference only, and are averaged over 10 inference calls after 5 warmup calls. Inference runs on an NVIDIA RTX 4090 GPU using bfloat16 precision and $n = 5$ denoising steps. BID [39] slows down inference due to sampling batches of actions, whereas RTC slows down inference due to backpropagating through each denoising step. BID (no forward contrast) refers to a version of BID without the forward contrast loss, which elides the need for a second model. BID (shared backbone) refers to a version of BID optimized specifically for the $\pi_0$ architecture, where the VLM backbone (3B parameters) is shared between the strong and weak model, so only two copies of the action expert (300M parameters) are needed. Full BID requires two copies of the entire model.

| Component | Time (mobile) | Time (non-mobile) |
|---|---|---|
| Model | 96.89 ± 0.16ms | 97.43 ± 0.28ms |
| Network | 21.20 ± 3.12ms | 6.89 ± 2.39ms |
| Image resize | 11.22 ± 5.00ms | 1.44 ± 0.27ms |
| Other | 9.67 ± 3.20ms | 3.00 ± 0.68ms |
| Total | 138.98 ± 6.71ms | 108.76 ± 2.34ms |

Table 2: Breakdown of **total** inference latency by component for RTC. The image resizing component happens on the CPU of the robot computer. In the mobile manipulation case, this computer is an Intel NUC portable computer with a 12th Gen Intel i7-1260P processor. In the non-mobile case, this computer is a desktop workstation with an AMD Ryzen 9 7950X processor. In both cases, the model runs on a separate workstation with an NVIDIA RTX 4090 GPU; the robot computer and the inference workstation are both connected to the same LAN via a wired Ethernet connection, and communication happens via the WebSocket protocol. Model inference uses bfloat16 precision and $n = 5$ denoising steps. Measurements are taken from 50 inference calls during a real episode rollout, and ± one standard deviation is shown.

| Component | Time (no RTC) | Time (with RTC) |
|---|---|---|
| Image encoders (SigLIP) | 18ms | 18ms |
| LLM prefill (Gemma 2B) | 44ms | 44ms |
| Denoising step (x5) | 14ms | 35ms |
| Total | 76ms | 97ms |

Table 3: Breakdown of **model** inference latency by component for vanilla $\pi_{0.5}$ and RTC. Measurements are taken from a single profiling trace for each method, run on an RTX 4090 GPU. RTC incurs a 2.5x latency increase per denoising step.

## A.4 Additional Simulated Ablations

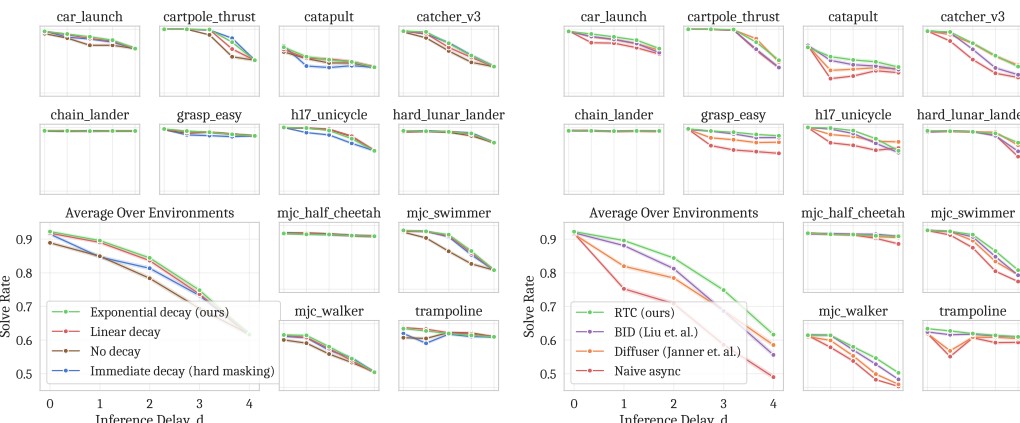

Figure 8: **Left:** Simulated ablation over different schedules for soft masking weights (Eq. 5). Exponential decay performs the best overall, although linear decay is very close behind. **Right:** Comparison with the inpainting algorithm from Diffuser [26], which overwrites a portion of the action chunk with the desired actions at each denoising step. While this simpler (and cheaper) inpainting method still provides some benefit, it is outperformed by our guidance-based approach.

## A.5 Hyperparameters

| Hyperparameter | Description | Simulation | Real-world |
|---|---|---|---|
| $n$ | Denoising steps | 5 | 5 |
| $H$ | Prediction horizon | 8 | 50 |
| $s_{min}$ | Minimum execution horizon | - | 25 |
| $\beta$ | Guidance weight clipping | 5 | 5 |
| $b$ | Delay buffer size | - | 10 |

Table 4: Hyperparameters used for RTC (Algorithm 1). In simulation, $d$ is held constant for each experiment, so $s_{min}$ and $b$ are not needed. Additional hyperparameters for the simulated experiments can be found in the code.

## A.6 Code Release

The code for the simulated experiments is available at `https://github.com/Physical-Intelligence/real-time-chunking-kinetix`.

## A.7 Compute Resources

All the experiments in this work use no more than 8 NVIDIA H100 GPUs (one NVIDIA DGX server) at a time. H100s are used via a cloud provider.

**Simulated experiments.** Training expert policies with RPO [50] with 6 seeds $\times$ 12 environments takes approximately 4 hours on 4xH100s. Generating data from those policies takes approximately 20 minutes on 6xH100s. Training imitation learning policies with flow matching for each environment takes approximately 1.5 hours on 2xH100s. Evaluating the policies for 2048 trials per environment takes approximately 5 minutes on 6xH100s.

**Real-world experiments.** We use policies fine-tuned from the $\pi_{0.5}$ [24] base model. Each fine-tuning run takes approximately 24 hours on 8xH100s. All of our real-world inference is done on a single NVIDIA RTX 4090 GPU in a workstation in the same building as the robots.

