# OpenReview forum: "Real-Time Execution of Action Chunking Flow Policies"
_NeurIPS.cc/2025/Conference — NeurIPS 2025 poster_

### Official Review · Reviewer_QwTe · 2025-06-30

**Clarity:** 3
**Significance:** 3
**Originality:** 3
**Rating:** 5
**Confidence:** 3

**Summary:**

This paper presents Real-Time Chunking (RTC), a novel inference-time algorithm that enables smooth asynchronous execution of action chunking policies for robot control. The key innovation is framing real-time execution as an inpainting problem, where the next action chunk is generated while executing the current one, "freezing" actions guaranteed to execute and "inpainting" the rest. The method is applicable to any diffusion- or flow-based vision-language-action (VLA) model without retraining. The authors validate RTC on 12 dynamic simulation tasks in Kinetix and 6 real-world bimanual manipulation tasks using the π0.5 VLA, demonstrating improved performance and robustness to inference delays up to 300ms.

**Questions:**

Can you provide detailed timing breakdowns comparing RTC to standard inference? How does the overhead scale with action dimension, prediction horizon, and number of denoising steps?

Why do RTC and hard masking perform similarly at d=0 and d=4 in Figure 5? What does this tell us about when soft masking is beneficial?

Under what specific conditions does RTC fail or perform poorly? Are there particular motion patterns, task characteristics, or environmental factors that challenge the inpainting approach?

What are the fundamental barriers to applying RTC to autoregressive VLAs? Have you explored any modifications that could extend the approach beyond diffusion/flow models?

**Ethical Concerns:**

["NO or VERY MINOR ethics concerns only"]

**Final Justification:**

I am generally satisfied with the authors' response, which has addressed my initial concerns. I am willing to maintain my accept rating to this paper.

**Limitations:**

The authors provide a reasonable discussion of limitations in Section 6, acknowledging the architectural constraints and computational overhead. However, the discussion would benefit from more in-depth analysis of computational costs.

**Paper Formatting Concerns:**

No major issues observed.

**Quality:**

3

**Strengths And Weaknesses:**

**Strength:**

This work addresses a practical and timely challenge: how to deploy powerful but slow neural policies in real-time robotic control. As large VLAs and diffusion-based controllers become more common, the latency issue could hinder their real-world use. RTC provides a solution that could be widely adopted to improve responsiveness.

The inpainting approach is intuitive and well-motivated. The soft masking extension (Section 4.2) that uses exponentially decaying weights for cross-chunk continuity is a clever contribution that improves upon naive hard masking.

The paper includes both simulated benchmarks (12 dynamic tasks) and real-world experiments (6 bimanual manipulation tasks), showing the effectiveness of the model in multiple scenarios.

The paper is well-written with effective visualizations that clearly illustrate the problem and solution.

**Weakness:**

While the paper acknowledges the computational cost of the inpainting procedure, it lacks detailed analysis.

In Figure 5, RTC and RTC with hard masking show similar performance when inference delay is zero or high (d=4). Why does soft masking provide no benefit in these cases? This pattern suggests potential limitations of the approach that are not adequately discussed.

 The paper provides limited discussion of failure modes or scenarios where RTC might struggle. Understanding when and why the method fails would help practitioners decide when to apply it and guide future improvements.

The comparison set is relatively narrow. While temporal ensembling and BID are relevant baselines, the paper would benefit from comparisons to other real-time execution strategies such as the hierarchical VLA designs mentioned in the related work.

---

> ### Author Rebuttal · Authors · 2025-07-30
>
> Thank you for your in-depth review! We have attempted to address your concern about computational costs by providing more details, which we will also add to the Appendix in a future revision (in addition to the existing latency breakdown in the Appendix). We have also provided some clarifications about the experiments. Please let us know if you have any other concerns!
>
> > In Figure 5, RTC and RTC with hard masking show similar performance when inference delay is zero or high (d=4). Why does soft masking provide no benefit in these cases?
>
> When $d = 0$, hard masking is equivalent to disabling RTC entirely, which performs well because there is no delay. When $d = 4$, hard masking is equivalent to soft masking, because the size of the region where soft masking would normally apply is $H - s - d = 0$.
>
> > The paper provides limited discussion of failure modes or scenarios where RTC might struggle. Understanding when and why the method fails would help practitioners decide when to apply it and guide future improvements.
>
> Across our simulated and real tasks, we did not find scenarios where RTC struggles (compared to synchronous baselines), so we would be confident recommending it as a reasonable default strategy.
>
> > Can you provide detailed timing breakdowns comparing RTC to standard inference? How does the overhead scale with action dimension, prediction horizon, and number of denoising steps?
>
> The RTC overhead comprises an approximate 50% slowdown of each denoising step. Thus, the overhead is linear in the number of denoising steps, independent of action dimension and prediction horizon. The total overhead depends on the model architecture — in Pi05, with 5 denoising steps, the denoising loop is approximately 20% of the total inference time without RTC, and 30% with RTC. We will add these details to the Appendix, as well as a more detailed table breaking down how much latency each component of the Pi05 architecture contributes (with and without RTC).
>
> > What are the fundamental barriers to applying RTC to autoregressive VLAs? Have you explored any modifications that could extend the approach beyond diffusion/flow models?
>
> The fundamental barrier is the lack of inpainting ability. If the actions were represented in standard chronological order, then one could simply “prompt” the model with the first $d$ frozen actions. However, [1] found that this standard action representation does not work well for high-frequency control, and their solution foils a naive prompting strategy. It is possible that alternative inpainting methods for autoregressive models exist, but we have not explored this direction yet.
>
> [1] “FAST: Efficient Action Tokenization for Vision-Language-Action Models.” Pertsch et al.

---

> > ### Comment · Reviewer_QwTe · 2025-08-08
> >
> > I thank the authors for their detailed rebuttal, which has successfully addressed my initial concerns. I will maintain my original rating.

---

### Official Review · Reviewer_rbym · 2025-07-03

**Clarity:** 4
**Significance:** 3
**Originality:** 2
**Rating:** 5
**Confidence:** 4

**Summary:**

This paper address a practical tension between the unavoidable inference latency of robot foundation models and the desired real-time control on real hardware.

---

## Problem Setting
The authors illustrate that for existing VLA models, the first few time steps in the produced action sequence has passed when the inference is done. The deviation in initial condition can cause jerky motions or completely off trajectories depending on the underlying controller.

Key assumptions:
- Builds from a trained diffusion or flow model that prediction action chunks.
- Asynchronous execution: the algorithm generates the next action chunk while the current one is still being executed.

---

## Method: Real-Time Chunking (RTC)

- The high level idea is to pose asynchronous action chunking as an inpainting problem.
- RTC aims to force the first $d$ steps from the new chunk (those executed during the inference delay) to match the corresponding actions from the previous chunk, and then inpaints the rest of the new chunk.
- This is done by adding a guidance term to the velocity field of the VLA. The guidance term can be computed with back propagation.
- The authors also propose soft masking: steps past the first $d$ also get exponentially decaying weights to match the previous chunk.

---

## Experiments
- The authors propose a suite of simulated task using force control. These tasks help highlight the effect of inference delay.
- In these simulated tasks, RTC is the most robust to inference delay - the task success rate drops the slowest as the inference delay increases. It also performs the best at different execution horizons.
- In real robot experiments, RTC achieved the highest task throughput across different levels of injected delay. Moreover, unlike the baselines, there is no performance drop as the delay increases.

**Questions:**

- Why does soft masking match the first $d$ steps better than hard masking? This is a bit counter-intuitive. Doesn't soft masking add more constraints? Also, because soft masking is encouraging closer match, why wouldn't it end up slowing down task progress?
- Would other types of low level controllers, such as more compliant controllers benefit more or less from this method?

**Ethical Concerns:**

["NO or VERY MINOR ethics concerns only"]

**Final Justification:**

I don't have major concern with the paper as a practical fix for flow-based VLA models with empirical justifications. I'll keep my original rating.

**Limitations:**

Yes.

**Quality:**

4

**Strengths And Weaknesses:**

## Strengths:
- No training required.
- This work addresses a timely and realistic issue with VLAs. The paper provides clear illustrations and real robot rollouts to motivate the problem.
- The experiment results are strong, showing that the method is able to avoid performance degradation at 200ms injected delay, more than the typical inference delay.

---

## Weaknesses:
- Limited to flow or diffusion models.
- This work applies an existing algorithm from image generation to robotics and falls short on novelty.
- Innovations like soft masking seems to work well empirically but lacks mathematical justification.

---

> ### Author Rebuttal · Authors · 2025-07-30
>
> Thank you for your in-depth review! We appreciate your acknowledgement of the timeliness and practicality of our problem setting. Regarding novelty, we would emphasize that we are the first work to study this problem setting (real-time execution with VLAs when the model latency exceeds a single controller timestep), that our baselines have not been applied to this problem setting before, and that we still provide an effective inference-only solution that outperforms these baselines.
>
> We have attempted to answer your questions about soft masking, and we will revise the paper to clarify these points, namely its role as an empirical (but not theoretically-motivated) modification to the existing inpainting algorithm. Please let us know if you have any other concerns!
>
> > Innovations like soft masking seems to work well empirically but lacks mathematical justification.
>
> That is a valid point, and we will add it to our limitations section.
>
> > Why does soft masking match the first $d$ steps better than hard masking?
>
> In the paper, we mention that “a small $d$ leads to a weak guidance signal and approximation errors can still cause discontinuities” (line 195). What we mean is that the inpainting algorithm is fundamentally a best-effort approximation, and in practice, we found that only using the first $d$ steps as guidance led to a poor match for small $d$. As you pointed out, this is theoretically unsatisfying, but it is an empirical reality.
>
> > Also, because soft masking is encouraging closer match, why wouldn't it end up slowing down task progress?
>
> The level of matching is a spectrum: from no matching at all, which is so bad that it is unusable on a real robot (see Figure 2, Figure 5, and the supplemental video), to perfectly matching a very long chunk of future actions, which could slow down task progress by preventing closed-loop correction. The right level of matching is an empirical question, but we found that with existing VLA hyperparameters (an action horizon of ~1 second), our soft masking approach with exponential decay performs well.
>
> > Would other types of low level controllers, such as more compliant controllers benefit more or less from this method?
>
> In theory, a more compliant controller could be more amenable to real-time execution, as the impact of sudden discontinuities between chunks is lessened. However, without RTC, the controller would merely provide a smoothing effect similar to that of temporal ensembling, which we have shown performs very poorly both in simulation and in reality.

---

> ### Comment · Reviewer_rbym · 2025-08-06
>
> I appreciate the authors’ efforts in addressing my comments from the original review.
>
> Overall, I find the paper to be a solid empirical study. I encourage the authors to expand their discussion on soft masking, including any alternative design choices they explored during development.

---

### Official Review · Reviewer_f8Dc · 2025-07-03

**Clarity:** 3
**Significance:** 4
**Originality:** 3
**Rating:** 5
**Confidence:** 4

**Summary:**

This paper presents a novel inference-time algorithm that enables smooth asynchronous execution of action chunking policies. The proposed method real-time chunking (RTC) is applicable to any diffusion- or flow-based VLA out of the box with no re-training. To test RTC, the authors introduce a new benchmark of 12 highly dynamic tasks in the Kinetix simulator, as well as evaluate 6 challenging real-world bimanual manipulation tasks. Results demonstrate favorable performance.

**Questions:**

1. How robust is the proposed method to sudden perturbations?
2. How sensitive is the overall system performance to the choice of inpainting algorithm?

**Ethical Concerns:**

["NO or VERY MINOR ethics concerns only"]

**Final Justification:**

I appreciate the authors’ efforts in addressing the concerns raised in my original review. I have also carefully considered the comments provided by the other reviewers.

With respect to my own feedback, I find that the authors have satisfactorily addressed the key issues in their response. Please ensure that the additional results mentioned are incorporated into the final manuscript.

Regarding the comments from Reviewer jJyz, I agree that the manuscript would benefit from a more comprehensive discussion of the existing literature in control, particularly with respect to explaining why those approaches are not applicable to the present work. Additionally, the connection and relevance of this work to cascade control systems should be clarified. Thus, I suggest the authors to tone down sentences like "this is a machine learning paper in submission at a machine learning conference, not a controls conference."

While the manuscript lacks a thorough discussion of the control literature, I believe the work maintains a clear focus on addressing a specific problem, i.e., executing action chunking using diffusion policies. Overall, I find it to be a solid empirical study with demonstrated performance, and I recommend that the paper be accepted.

**Limitations:**

Yes.

**Quality:**

3

**Strengths And Weaknesses:**

Strengths:
1. Timely Contribution to Real-Time Control with Large Models: The application of large models to real-time control is a topic of growing importance in the robot learning community, especially given the inherent latency and computational demands of such models. Addressing this challenge requires effective asynchronous mechanisms. The authors’ proposed method, RTC, is a compelling step in this direction. Figure 1 clearly illustrates RTC’s ability to generate significantly smoother trajectories compared to existing baselines, highlighting its practical benefit. Furthermore, the accompanying demonstration video effectively showcases RTC’s real-time responsiveness and stability in dynamic scenarios.

2. Strong Empirical Validation Across Simulated and Real-World Settings:
The authors introduce a new benchmark built on the Kinetix simulator, featuring 12 diverse and highly dynamic tasks spanning both manipulation and locomotion. This benchmark offers a meaningful testbed for evaluating real-time performance under challenging conditions. In addition, the real-world experiments—conducted on 6 complex bimanual manipulation tasks using the π₀.₅ VLA as the base policy—provide strong empirical support for the effectiveness of RTC. Across both domains, the results consistently show that RTC achieves high performance with low latency, underscoring its practical viability for real-time deployment.

Weaknesses:
1. Clarifying the Broader Impact of Computational Overhead: This work presents a compelling method for enabling real-time control using large models, supported by strong empirical results. However, in the limitations section, the authors primarily acknowledge the added computational overhead introduced by their method. I encourage the authors to further examine and report on whether this overhead affects other critical aspects of system performance—such as latency variability, responsiveness to dynamic changes, or robustness under constrained compute conditions. Understanding the broader implications of this overhead is essential for assessing the method’s practical deployment in resource-limited settings, such as on embedded platforms or mobile robots. One common evaluation scenario is testing robustness to sudden perturbations. It would be valuable to understand how the proposed method performs under such conditions and whether the increased computational cost compromises its responsiveness or stability.

2. Role and Generality of the Inpainting Algorithm: The proposed framework relies heavily on an inpainting algorithm to recover missing observations, which plays a crucial role in enabling effective asynchronous control. In this study, the authors adopt the method by Pokle et al. [46]. However, it remains unclear how sensitive the overall system performance is to this choice. Could alternative inpainting algorithms lead to better, worse, or more efficient outcomes? A comparison or ablation study exploring different inpainting strategies would greatly enhance the generality and reproducibility of the proposed approach. It would also clarify whether the strength of the results stems primarily from the RTC framework itself or is heavily dependent on the chosen inpainting technique.

---

> ### Author Rebuttal · Authors · 2025-07-30
>
> Thank you for your in-depth review! We have attempted to address your question about inpainting strategies by adding a simulated ablation: it shows that RTC is still valuable with simpler inpainting strategies, but the work we have done to adapt and tune a state-of-the-art method is critical for maximum performance. Additionally, we have clarified the impact of additional computational overhead and sudden perturbations. Please let us know if you have any other concerns!
>
> > However, in the limitations section, the authors primarily acknowledge the added computational overhead introduced by their method. I encourage the authors to further examine and report on whether this overhead affects other critical aspects of system performance—such as latency variability, responsiveness to dynamic changes, or robustness under constrained compute conditions.
>
> In Table 2 of the Appendix, we report mean and standard deviations of inference times for each component of the system. We find that even with RTC, latency variance for model inference is very low (<1ms); most of the variance comes from the network component, which is unrelated to RTC. Regarding responsiveness, RTC is necessary to enable responsiveness under delays (see our answer below), so it is difficult to divorce the negative impact of the additional computational overhead from the positive impact of the method itself. Regarding constrained compute conditions, RTC is targeted primarily at offboard inference scenarios, where compute is plentiful, but the network adds inevitable latency. However, if one were to use RTC in onboard inference, we do not see any reason why the backpropagation in RTC would disproportionately affect a smaller GPU (the relative overhead would be the same as on a powerful GPU).
>
> > How robust is the proposed method to sudden perturbations?
>
> Unfortunately, we cannot run more real-world experiments, but we can point out that in the simulated benchmark, the actions are perturbed with random Gaussian noise at every step (lines 231-232). Our results (Figure 5) show that RTC is more robust than other methods, and that it strictly benefits from increasingly closed-loop control (Figure 5, bottom left).
>
> > A comparison or ablation study exploring different inpainting strategies would greatly enhance the generality and reproducibility of the proposed approach.
>
> We have performed an additional simulated ablation comparing to the inpainting strategy from Diffuser [1], which is simpler and does not include a gradient-based correction. Unfortunately, we cannot attach images to the rebuttal, but the inpainting strategy from Diffuser performs worse than standard RTC across the board; it performs worse than BID at $d = 1$ and $d = 2$, and better than BID at $d = 3$ and $d = 4$; and it performs much better than naive asynchronous and temporal ensembling. Our conclusion is that our full method — with a state-of-the-art inpainting algorithm and soft masking — is important for maximum performance, but that a simpler inpainting strategy can still provide some benefit.
>
>
>
> [1] “Planning with Diffusion for Flexible Behavior Synthesis.” Janner et al.

---

> ### Comment · Reviewer_f8Dc · 2025-08-07
>
> I appreciate the authors’ efforts in addressing the concerns raised in my original review. I have also carefully considered the comments provided by the other reviewers.
>
> With respect to my own feedback, I find that the authors have satisfactorily addressed the key issues in their response. Please ensure that the additional results mentioned are incorporated into the final manuscript.
>
> Regarding the comments from Reviewer jJyz, I agree that the manuscript would benefit from a more comprehensive discussion of the existing literature in control, particularly with respect to explaining why those approaches are not applicable to the present work. Additionally, the connection and relevance of this work to cascade control systems should be clarified. Thus, I suggest the authors to tone down sentences like "this is a machine learning paper in submission at a machine learning conference, not a controls conference."
>
> While the manuscript lacks a thorough discussion of the control literature, I believe the work maintains a clear focus on addressing a specific problem, i.e., executing action chunking using diffusion policies. Overall, I find it to be a solid empirical study with demonstrated performance, and I recommend that the paper be accepted.

---

> ### Author Response · Authors · 2025-08-08
>
> Thank you for your response! We will certainly include these additional ablations in the paper, and we appreciate your help in improving this work.
>
> Regarding the field of control, we want to clarify that we in no way disagree that the paper would be improved by a more thorough treatment of existing literature in control, which includes many closely related ideas. We actually had a more detailed discussion of MPC in an initial draft, given its similarities with action chunking. However, we decided to cut it in favor of more thorough discussion of learning-centric related work, which we thought would be more important for the NeurIPS audience. With the additional space afforded by the camera-ready, we can certainly put this back, and clarify why these methods are not directly applicable to our problem setting (essentially, elaborating on lines 96-97: "their reliance on explicit dynamics models and cost functions makes their application difficult in unstructured settings..."). We agree that discussion of cascade control is missing, and that discussing its relationship with RTC -- and action chunking more broadly -- would also be beneficial.
>
> We apologize if our response to reviewer jJyz came across aggressively -- we were specifically responding to their comment that "in a well-understood field of control (rather than diffusion based trajectory generation), more care needs to be taken." We merely wanted to highlight that we did not intend for this to be framed as a "controls paper" per se, and that we fully agree that our work is limited to diffusion-based action generation. We also meant to highlight that the venue (NeurIPS) influenced our writing and framing. We hope that is clear from the manuscript -- especially the introduction, where we deliberately tried to motivate the problem first from a general large-scale deep learning perspective, then from a robot learning perspective. We would welcome any feedback in clarifying the motivation further!

---

### Official Review · Reviewer_jJyz · 2025-07-03

**Clarity:** 3
**Significance:** 2
**Originality:** 2
**Rating:** 2
**Confidence:** 5

**Summary:**

This paper address the problem of how to accelerate the execution of policies encoded using action chunking. The approach uses conditional flow matching, and rather than perform synchronous action chunk generation, where the next action chunk is generated after the previous action chunk completes, the approach generates the next action chunk during the previous action chunk execution, and uses inpainting to ensure that the generated action chunk is consistent with the action chunk being generated. The experimental results show that in some cases the execution of the full policy is faster and is sometimes able to complete the task more reliably.

**Questions:**

- Why does the performance drop at d = 4? Why do the results not go to d = 6?
- What happens in this paper when the flow matching returns quickly?
- How sensitive is the performance of the model to changes in the exponential weight decay function?

**Ethical Concerns:**

["NO or VERY MINOR ethics concerns only"]

**Final Justification:**

I believe that the authors have made a good faith effort to address my concerns. I am willing to overlook the unfortunate response, which the authors did attempt to walk back. However, my original criticism stands, and I am not supportive of the paper.  I really wish that the authors had taken the opportunity to take a step back at the onset of the research, and identify what is already known about problems where the system will evolve out from under the decision making system. This is a well-studied problem and the authors do themselves and the community a disservice by not doing that kind of scholarship first.

I understand that the specific problem of how to deal with this in the context of diffusion, and the effect of smoothed inpainting, may not have been tried before (it would be hard to imagine it had been tried before, given how new diffusion is as a practical tool). But, I am arguing for going beyond simply trying something and reporting its results, however carefully analysed those results are.

**Limitations:**

The limitations section is exceedingly short. The authors should be encouraged to put more careful analysis into their technique.

**Quality:**

2

**Strengths And Weaknesses:**

The problem, as posed, is fairly clear and relevant, and the paper is reasonably clearly written and the technique is sound.

The primary difficulties I have with this paper are the problem setting and the experimental evaluation. Firstly, the paper describes the proposed approach as real-time, but real-time has a specific meaning, which is that there is a guaranteed response time. This paper accelerates execution but there is no discussion of a guarantee on response time, or a response time to what. It is not clear it is possible to guarantee that a flow matching technique can be made real time, in that if a given denoising step takes longer than the required response time for whatever reason, the policy cannot return early. This might seem like a quibble to the authors and can probably be addressed by changing the title and framing of the paper, but it is an important point in how the authors take about their work.

The lack of clarity also shows up in the experimental results. The authors describe a minimum delay of somewhere between 76 and 98ms, plus 25ms of transmission time over the LAN, for a minimum of $d \approx 6ms$. However, we see in figure 5 that the performance drops substantially up to $d = 4$, and results are not given beyond that. The authors then state "we would like to understand how the system behaves with higher inference latencies," and inject latencies of 100ms and 200ms, which are large (arguably unrealistic?) latencies given the stated properties of the existing system.

The authors are not especially careful in presenting their results. For instance, "we find that hard masking somewhat underperforms soft masking, particularly when d is smaller." The fact that there are no error bars raises the question of how significant this is, but the authors need to be careful to point out that hard masking *overperforms* soft masking when d is large in the cartpole thrust problem. The text of the paper barely mentions that in both figure 5 and figure 7, we see that no technique is able to solve the problems with a high rate of success (solve rate is not defined. I assume this is the proportion of trials where the task was completed correctly.), and the differences are relatively modest -- naive async is 70%ish vs RTC at about 78 or 79% averaged across sim environments. For the real results, only for dishs_in_sink and shirt_folding do we see what appears to be 100% completion (the 100% line is not labelled). And for dishes_in_sink, only TE seems to achieve completion.

The approach has some parameters, and it would be helpful to know how sensitive performance is to the choice of parameters (number of denoising steps, faster denoising vs open-loop performance. Equation 5 gives the exponential decay of the weights during in-painting -- if the exponential were encouraged to delay faster, or more slowly, how sensitive is the performance of the model?

The authors are also not rigorous in the related work in terms of MPC techniques -- the issue of executing a partial control and then generating a new one is a very well-studied problem in control -- there are principled techniques from people like Chris Gerdes who have examined when to to switch to a new control given a partial plan. What happens in this paper when the flow matching returns quickly? This can lead to thrashing, and is not discussed in this paper.

I dislike the term incremental, in that incremental results are important, but the relative modest technical contribution of this paper puts a lot of onus on careful experimental evaluation, which is not present in this paper.

---

> ### Author Rebuttal · Authors · 2025-07-30
>
> Thank you for your in-depth review! We agree that framing our claims properly is very important, and we appreciate your help in treating this subject carefully. We will attempt to address your concerns by: (1) expanding and clarifying our usage of the term “real-time”, and being more explicit about our assumptions; (2) adding more simulated ablations; and (3) clarifying a number of points that we believe may stem from misunderstandings, regarding which we will improve intelligibility in a future revision. Please let us know if any further issues remain!
>
> > Firstly, the paper describes the proposed approach as real-time, but real-time has a specific meaning, which is that there is a guaranteed response time. This paper accelerates execution but there is no discussion of a guarantee on response time, or a response time to what.
>
> This is a great point! We had a discussion among the authors about this exact point, and ultimately decided on “real-time” for the following reason: we can think of each low-level controller timestep as an “event”, and the procurement of the next action as the “response”. Prior synchronous action chunking strategies are not real-time because, every $s$ steps (the execution horizon), the controller pauses and waits for model inference, meaning the event-response time is suddenly and unexpectedly many times higher. In RTC, every response comes immediately (in $< \Delta t$ seconds) because model inference is done asynchronously with controller execution.
>
> We agree that this detail is important, and we will update the paper to explain how RTC satisfies the real-time guarantee, namely that it always returns an action response within the $\Delta t$ deadline. As with any real-time system, it is possible for this guarantee to fail in cases of catastrophic failure, e.g., a temporary network outage that causes $d$ to exceed $H - s$ (this never happened throughout our 480 evaluation episodes). However, we will also clearly state this possibility as a limitation.
>
> > The authors describe a minimum delay of somewhere between 76 and 98ms, plus 25ms of transmission time over the LAN, for a minimum of $d \approx 6ms$. However, we see in figure 5 that the performance drops substantially up to $d = 4$, and results are not given beyond that.
>
> We believe that there are a few misunderstandings here, and we will clarify the writing on this point in a future revision. 76ms is the model inference delay for the baselines, and 98ms is the model inference delay for RTC (line 263). $d$ is in units of controller steps, not milliseconds, and is calculated using the equation on line 129: $d = \lfloor(98ms + 25ms) / (20ms)\rfloor = 6$. We will expand that section to clarify the distinction.
>
> Figure 5 describes the simulated experiments, not real-world experiments, meaning that we control $d$ directly independent of model latency. $d = 4$ is the maximum because we train with a horizon of $H = 8$, and the maximum possible $d$ is $\lfloor H / 2 \rfloor$ (mentioned on line 238). This means that our simulated experiments test very high relative latencies — up to half of the prediction horizon — which is why performance drops so much across the board (but less so for RTC).
>
> > and inject latencies of 100ms and 200ms, which are large (arguably unrealistic?) latencies given the stated properties of the existing system
>
> These are very realistic latencies. Many VLAs are already much slower than Pi05: “Kim et al. [30], who optimize the 7B OpenVLA model [29] specifically for inference speed, achieve no better than 321ms of latency on an NVIDIA A100 GPU” (line 145). If the scaling properties from vision and language hold in robotics, we should expect the best-performing VLAs to increase in size as time goes on (mentioned in line 37). Additionally, as mentioned throughout the paper, our experiments were conducted in ideal network conditions with powerful workstations; weaker devices and/or slow networks could easily add latency in the hundreds of milliseconds. These justifications for studying injected latency are mentioned on lines 265-266 of the paper.
>
> > For instance, "we find that hard masking somewhat underperforms soft masking, particularly when d is smaller." The fact that there are no error bars raises the question of how significant this is
>
> Figure 5, which compares hard and soft masking in the simulated benchmark, has 95% Wilson score intervals shaded in (as stated in the figure caption).
>
>
> > solve rate is not defined. I assume this is the proportion of trials where the task was completed correctly
>
> The Figure 5 caption mentions that each Kinetix environment “involves getting a green object on the left to touch a blue one on the right.” This is what counts as a “success”; we will update the Figure 5 caption to clarify this point and explicitly define the solve rate.
>
> > Equation 5 gives the exponential decay of the weights during in-painting -- if the exponential were encouraged to delay faster, or more slowly, how sensitive is the performance of the model?
>
> We have performed an additional simulated ablation that we will add to the paper (unfortunately, the rebuttal response does not allow images). We compared 4 schedules: exponential decay (our default), linear decay, constant ones (no decay), and constant zeros (hard masking; already in the paper). Averaged over environments, exponential performs the best, followed by linear, followed by hard masking, followed by no decay. Exponential and linear both outperform BID by a large margin.
>
>
>
> > The authors are also not rigorous in the related work in terms of MPC techniques -- the issue of executing a partial control and then generating a new one is a very well-studied problem in control -- there are principled techniques from people like Chris Gerdes who have examined when to to switch to a new control given a partial plan
>
> Could you point us to some more specific references? We are not aware of any particular techniques from the MPC literature that would be applicable out-of-the-box to our setting, but we would be happy to add any relevant references.
>
> > What happens in this paper when the flow matching returns quickly? This can lead to thrashing, and is not discussed in this paper.
>
> How often the chunk changes can be controlled independently of how quickly the flow matching returns, and is controlled by the hyperperameter $s_\text{min}$ (see Algorithm 1, line 13, as well as lines 210-212 of the text). If the flow matching returns quickly, it allows for a lower execution horizon, but does not require it (see line 211 of the text). However, RTC can take advantage of a lower execution horizon due to its consistency across chunks (Figure 5, lower left).
>
> If by “thrashing”, you are referring to the phenomenon where the robot switches between different plans too quickly, this problem is discussed throughout the paper and is at the core of the motivation for RTC (lines 45, 154-164, 253-255, Figure 2). Apologies if we are misunderstanding your terminology.

---

> > ### Comment · Reviewer_jJyz · 2025-08-06
> >
> > Thanks to the authors for their detailed response.
> >
> > The authors are correct that I had confused d as an absolute delay in ms, rather than an number of time steps, and also addresses my comment about "100ms and 200ms which are large (arguably unrealistic?) latencies", since the experiments appeared (in my reading) to be injecting larger latencies than what were being experienced. So, happy to retract those comments.
> >
> > That being said, I am still not supportive of accepting this paper, and my overall point is that the authors are not especially careful about presenting their results, even with some of their responses, and it is not clear to me how well this technique actually works. It also feels very much like they have a very specific problem related to diffusion model learning, and in a well-understood field of control (rather than diffusion based trajectory generation), more care needs to be taken to do proper framing of the problem and experimental evaluation.
> >
> > I do not agree that this paper addresses thrashing, or that it is at the core of the motivation. This paper deals with the problem of how to ensure that a new plan returned at a point in the future can be compatible with the old plan. From page 4: "The key challenge in real-time execution is to maintain continuity between chunks. By the time a new chunk is available, the previous one has already been executed partway, and therefore the new chunk must be “compatible” with the previous one." What I meant by "thrashing" is when an outer loop (e.g., the behaviour chunk diffusor) generates a new plan to be executed by an inner loop very quickly, oscillations can occur. This phenomenon is often encountered in cascade control. I tried to find an online reference, but didn't have great success because this is an older concept that is most often found in controls textbooks. The paper "Robust Analysis and PID Tuning of Cascade Control Systems" (Tan et al, 2005) seems a reasonable treatment and describe the phenomenon I am concerned about on 6 where that paper (by Tan et al) says "If the inner loop is fast compared to the outer loop, then the effect of one loop on the other loop is small." This phenomenon is exacerbated by noise between the loops. The solution in a cascade controller is usually to slow down the outer loop (artificially inducing latency), with a rule of thumb that the outer loop should run at about 1/10 of the inner loop. This is *not* what this paper under review is about, the paper states "A small d leads to a weak guidance signal" (which presumably means their technique has limited effect) and my question stands. I wonder if the latency is actually helping performance in ways the authors do not understand.
> >
> > In response to the question about more specific references, since my pointer to Chris Gerdes and MPC was not sufficient, then
> > John K. Subosits and J. Christian Gerdes. “From the Racetrack to the Road: Real-Time Trajectory Replanning for Autonomous Driving”, IEEE Transactions on Intelligent Vehicles, Vol. 4, No. 2, June 2019
> > and
> > NR Kapania, J Subosits, J Christian Gerdes. "A sequential two-step algorithm for fast generation of vehicle racing trajectories". Journal of Dynamic Systems, Measurement, and Control, 2016.

---

> > > ### Author Response · Authors · 2025-08-06
> > >
> > > > my overall point is that the authors are not especially careful about presenting their results, even with some of their responses, and it is not clear to me how well this technique actually works
> > >
> > > Now that the misunderstandings have been resolved, could you clarify in more detail where you think we were not careful in presenting our results, and why it is not clear that RTC actually works? We have shown statistically significant improvements over baselines in both simulation and real-world robotics. The real-world evaluation, in particular, includes 480 evaluation episodes across 6 diverse tasks adding up to 28 hours of pure robot execution time, plus post-hoc granular scoring. In the realm of real-time execution strategies, it can be observed from the supplemental video that at +100ms and +200ms of delay, temporal ensembling does not work at all, leading to such violent movements that the robot hardware fails within seconds, while RTC experiences no degradation in throughput.
> > >
> > > > It also feels very much like they have a very specific problem related to diffusion model learning, and in a well-understood field of control (rather than diffusion based trajectory generation), more care needs to be taken to do proper framing of the problem and experimental evaluation.
> > >
> > > We would agree that this paper does solve a specific problem related to executing action chunking diffusion policies. We would highlight that this is a machine learning paper in submission at a machine learning conference, not a controls conference. We stand by the novelty of our problem setting: namely, how to achieve real-time execution within the increasingly popular and patently effective diffusion-based action chunking paradigm. We stand by the quality of our experimental results: they are thorough and conclusive in showing that RTC is useful for real-world robotics. If there are obvious baselines inspired by the field of control that are in scope and directly applicable to robot learning that we missed, we would be happy to address these.
> > >
> > > > I do not agree that this paper addresses thrashing, or that it is at the core of the motivation
> > >
> > > Thank you for clarifying your usage of the term “thrashing”, and we believe that we did misunderstand your earlier comment. We thought you were making an analogy between thrashing and the problem of fast switching between incompatible plans, which can lead to large oscillations in certain cases, and is discussed in our work. If by thrashing, you are referring to something more specific related to cascaded feedback controllers, we believe this is out of scope for our paper since our paper is not specifically about cascaded control and is agnostic to the inner control loop. (For example, in our simulated experiments, there is no inner control loop since the policy outputs forces directly.) Many cases of action chunking can be viewed as cascaded control systems, in which case thrashing is a valid concern, but this criticism also applies to the vast quantity of recent papers in the robot learning literature that use action chunking.
> > >
> > > > the paper states "A small d leads to a weak guidance signal" (which presumably means their technique has limited effect)
> > >
> > > We believe there is another misunderstanding here. This quote is from Section 4.2, which describes the motivation for soft masking, and strengthening the guidance signal when d is small is part of this motivation. With soft masking, RTC is very effective, as illustrated qualitatively in Figure 4 (right next to Section 4.2) as well as quantitatively our experiments (even with small d; see Figure 5, bottom right). We will clarify the writing regarding soft masking in a future revision, as mentioned in our response to reviewer rbym.
> > >
> > > > In response to the question about more specific references, since my pointer to Chris Gerdes and MPC was not sufficient,
> > >
> > > We appreciate the specific references. We have reviewed them and are happy to add discussion of them to the related works section. The two papers you provided are specific to vehicle racing trajectories and use convex optimization in conjunction with hand-crafted vehicle dynamics models. These methods are not applicable to our setting, as we used a pre-trained action chunking policy trained on a corpus of demonstration data, with no dynamics modeling.

---

### Note · Authors · 2025-08-12

Dear AC and reviewers,

Thank you all for the feedback and discussion. We believe that we’ve addressed the majority of the issues raised by reviewers, adding the following quantitative results:

- A simulated ablation comparing 4 decay schedules: exponential decay (our default), linear decay, constant ones (no decay), and constant zeros (hard masking; already in the paper); the results show that exponential is the best.
- A simulated ablation comparing our method to the inpainting strategy from Diffuser (Janner et. al.). It performs better than naive asynchronous and temporal ensembling, but worse than the guidance-based inpainting approach.
- A more in-depth benchmark of the computational overhead from RTC, showing that it adds a 50% overhead for the denoising loop.

Our primary action items for revising the paper are as follows:

- We will define “real-time,” and explain how our method satisfies a real-time constraint: namely, that the system always produces an action every $\Delta t$ seconds. We will clarify that our method can fail to meet this real-time deadline in cases of catastrophic failure (e.g., a network outage) that cause $d$ to exceed $H - s$, although this never happened in our experiments.
- We will expand the related work section with a more thorough treatment of the controls literature, including MPC and cascade control (given their similarities to action chunking). We will clarify why prior MPC methods are not directly applicable to our problem setting (essentially, elaborating on lines 96-97: "their reliance on explicit dynamics models and cost functions makes their application difficult in unstructured settings..."). We will also explain why cascade control is not directly applicable to our setting: namely, that we are agnostic to the design of the low-level controller, or even the presence of a low-level control loop (e.g., our simulated benchmark has none).
- We will expand Section 4.2: Soft Masking to clarify that it is an empirically, not theoretically, motivated design decision. We will explain how it arose from a desire to obtain a better match for the first $d$ actions when $d$ is small, and that the exponential decay is based on the intuition that the policy should be allowed to deviate more as time goes on, balancing the desire for continuity between chunks with the desire for fast task progress. Here, we will also point to the additional simulated ablation showing that exponential decay performs the best.

---

### Decision · Program_Chairs · 2025-09-17

**Decision:**

Accept (poster)

**Comment:**

After carefully considering the mixed reviews and rebuttal, I find the contribution technically sound and impactful. RTC introduces a practical inference-time algorithm for real-time execution of diffusion/flow-based VLA policies, showing strong improvements across both simulation and real-world robotics tasks with robustness to latency. Positive reviews (f8Dc, rbym, QwTe) highlight the method’s timeliness, broad applicability without retraining, and convincing empirical validation. The main negative review (jJyz) raised concerns about framing of “real-time,” insufficient positioning in the control literature, and presentation care, though some initial misunderstandings were later clarified. For the camera-ready, I encourage the authors to: (i) revise the framing of “real-time” and clarify guarantees, (ii) expand discussion of control-related works as Reviewer jJyz emphasized, (iii) provide fuller analysis of computational overhead and trade-offs, and (iv) explicitly discuss limitations, sensitivity to inpainting strategies, and potential failure modes.